# A Questionnaire Integrated with the Digital Medical Record Improved the Coverage of a Control Program for Congenital Chagas Disease in Tuscany, Italy

**DOI:** 10.3390/microorganisms11010154

**Published:** 2023-01-06

**Authors:** Anna Barbiero, Martina Mazzi, Antonia Mantella, Michele Trotta, Gian Maria Rossolini, Alberto Antonelli, Patrizia Bordonaro, Maria Grazia Colao, Anna Rosa Speciale, Tullio Di Benedetto, Mariarosaria Di Tommaso, Elisabetta Mantengoli, Felice Petraglia, Luisa Galli, Marco Pezzati, Carlo Dani, Maria José Caldes Pinilla, Cecilia Berni, Bassam Dannaoui, Pedro Albajar Viñas, Alessandro Bartoloni, Lorenzo Zammarchi

**Affiliations:** 1Department of Experimental and Clinical Medicine, University of Florence, Largo Brambilla 3, 50134 Florence, Italy; 2School of Human Health Sciences, University of Florence, Largo Brambilla 3, 50134 Florence, Italy; 3Tuscany Regional Referral Center for Tropical Diseases, Careggi University Hospital, 50134 Florence, Italy; 4Infectious and Tropical Diseases Unit, Careggi University Hospital, Largo Brambilla 3, 50134 Florence, Italy; 5Tuscany Regional Referral Center for Infectious Diseases in Pregnancy, Careggi University Hospital, Largo Brambilla 3, 50134 Florence, Italy; 6Clinical Microbiology and Virology Unit, Careggi University Hospital, 50134 Florence, Italy; 7Hospital General Laboratory, Careggi University Hospital, Largo Brambilla 3, 50134 Florence, Italy; 8Department of Health Sciences, Obstetrics and Gynecology Branch, University of Florence, Largo Brambilla 3, 50134 Florence, Italy; 9Biomedical, Experimental and Clinical Sciences “Mario Serio”, University of Florence, Largo Brambilla 3, 50134 Florence, Italy; 10Department of Health Sciences University of Florence, Paediatric Infectious Diseases Division, Anna Meyer Children’s University Hospital, Viale Pieraccini 24, 50139 Florence, Italy; 11Pediatric Unit, Santa Maria Annunziata Hospital, AUSL Toscana Centro, Via Antella, 58, Bagno a Ripoli, 50012 Florence, Italy; 12Department of Neuroscience, Psychology, Drug Research and Child Health, Careggi University Hospital of Florence, Largo Brambilla 3, 50134 Florence, Italy; 13Tuscany Regional Center for Global Health, Anna Meyer Children’s University Hospital, Viale Pieraccini 24, 50139 Florence, Italy; 14Citizenship Rights and Social Cohesion Directorate, Tuscany Region, Via Camillo Cavour, 2, 50122 Florence, Italy; 15Technological Innovation in Clinical-Assistance Activities Unit, Azienda Ospedaliero-Universitaria Careggi, Largo Brambilla 3, 50134 Florence, Italy; 16Department of Control of Neglected Tropical Diseases, World Health Organization, Avenue Appia 20, CH-1211 Geneva, Switzerland

**Keywords:** pregnancy, *Trypanosoma cruzi*, neglected tropical disease, congenital, TORCH

## Abstract

The leading route of Chagas disease transmission in nonendemic countries is congenital. However, policies concerning screening, prevention, and management of congenital Chagas disease are rare in these settings. Since 2012, serological screening for Chagas disease should be provided for pregnant women at risk in Tuscany, Italy according to a Regional resolution. Due to difficulties in the implementation, in November 2019, a checklist aimed at identifying pregnant women at risk for Chagas disease was introduced in digital clinical records at Careggi University Hospital, Florence, Italy. In order to evaluate the effectiveness of the “Chagas checklist”, data about the number of deliveries by women at risk and their screening coverage between 2012 and June 2022 were collected. Out of 1348 deliveries by women at risk, 626 (47%) *Trypanosoma cruzi* serology tests were performed during the study period. The annual screening coverage increased from an average of 40.3% between 2012 and 2019 to 75.7% between 2020 and June 2022, underlining the big impact of the checklist. Four Chagas disease serological tests out of 626 (0.6%) resulted positive, corresponding to 2 affected women. No cases of congenital transmission occurred. The study showed that a simple digital tool led to a tangible improvement in the coverage of the screening program; its application in a setting where digital charts are available will contribute to the control and elimination of congenital Chagas disease.

## 1. Introduction

*Trypanosoma cruzi* infection, also known as Chagas disease, is an infectious parasitic disease that is endemic in 21 Latin American countries. Its distribution follows that of the main vector of the parasite, the triatomine bug. Infection in endemic countries mainly occurs through vectorial transmission after a bite from an infected triatomine bug, with deposition of *T. cruzi* contaminated feces on the skin of the human host and penetration of the parasite through skin or mucosae. More rarely, infection can occur through ingesting food or drinks contaminated by infected triatomine feces. *T. cruzi* infection can also occur through maternal–fetal transmission, blood transfusion, and organ transplant from infected donors [1,2].

It is estimated that 6 to 7 million people are affected worldwide by Chagas disease and 20–30% of them will likely develop cardiac or gastrointestinal typical complications of the disease [3]. The majority of infected people live in Latin American endemic countries; however, migration phenomena and restrictive migration policies on the part of the United States have resulted in a steady increase in the number of Latin American migrants in Europe in recent decades, especially to Spain and Italy leading to the diffusion of Chagas disease to nonendemic European countries. Indeed, although the vector is not present in nonendemic areas, congenital, blood-transfusion, or organ-transplant transmission can perpetuate and strengthen the burden of the disease [4,5].

Currently, the World Health Organization (WHO) estimates that up to 8664 infants are born with the disease each year, accounting for about 22.5% of new incident cases, with the congenital route of transmission being the second most important route after the vectorial one in endemic countries and the first way of transmission in nonendemic countries, where the underdiagnosis rate of disease is estimated to range between 94 and 96% [6,7]. While Chagas disease screening in blood and organ donors is quite widespread among European countries, screening programs for the prevention and control of congenital Chagas disease are still lacking in nonendemic countries [8,9,10]. The congenital transmission rate from a mother affected by *T. cruzi* infection to her child is highly variable but settles around 5% in endemic countries and 3.8% in nonendemic countries [11].

Infected newborns are asymptomatic in 60–90% of cases and have a 20–30% probability of developing chronic complications of the disease during life [12]. The symptomatic acute congenital infection can present with a broad range of symptoms such as low weight at birth, low Apgar score, hepatosplenomegaly and, with lower frequency, myocarditis with cardiovascular insufficiency, meningoencephalitis, and respiratory distress syndrome, with high mortality rate [12,13,14]. The most common obstetric complication in mothers affected by Chagas disease appears to be premature rupture of membranes [14].

Italy is the second country in Europe and third in the world for the number of Latin American immigrants. In 2020, 366,343 migrants from Central and Southern America were recorded in Italy, of which 60.9% (223,260) were women [15]. Considering seroprevalence in countries of origin, it is estimated that about 3000 to 5000 people are affected by chronic Chagas disease in Italy [16], with 2000 cases of cardiac complications and a 98–99% rate of underdiagnosis [17,18,19,20]. In Tuscany, 25,130 citizens coming from Central and Southern America were registered in 2020, of which 61.3% (15,397) were women. Within the female immigrant population from Latin America, 39.1% come from Peru, 17.2% from Brazil, 4.5% from Ecuador, and to a lesser extent from Colombia, Venezuela, Argentina, Mexico, Bolivia, Chile, and other endemic countries [15].

With the awareness that the burden of Chagas disease is still largely unknown in nonendemic countries [21] and that improvement of screening programs is needed, not only at the regional level but also at national and European levels, a first screening program for Chagas disease in pregnancy was implemented in Tuscany region in 2012.

In this work, we aim to describe the screening program implemented in Tuscany and to assess the level of the implementation and the epidemiological impact of the protocol in Careggi University Hospital between 2012, after its first introduction, and June 2022, comparing screening coverage rates before and after the introduction of an informatic reminder tool called the Chagas checklist.

## 2. Materials and Methods

### 2.1. Screening Program

With several Regional resolutions, issued in 2012, 2015, and 2019, respectively, (“Programma regionale per la prevenzione e il controllo della malattia di Chagas congenita”; “delibere n°489 04-06-2012, n°659 25-05-2015 and n°565 23-04-2019, Decreto Dirigenziale 21395/2019”) [22], the Tuscany region included the serological test for Chagas disease (*T. cruzi* IgG serology), among the free laboratory investigations that should be routinely performed during pregnancy, in addition to the serological tests recommended by Italian national law (which include toxoplasmosis, rubella, syphilis, HIV, HCV, HBV serologies) according to a National decree published on 12 January 2017 [23].

Women at risk for Chagas disease are defined as those born in Latin America endemic countries (Argentina, Belize, Bolivia, Brazil, Chile, Colombia, Costa Rica, Ecuador, El Salvador, French Guyana, Guatemala, Guyana, Honduras, Mexico, Nicaragua, Panama, Paraguay, Peru, Suriname, Uruguay, Venezuela), or women born to mothers from those countries [22]. The main objectives of the screening program include serological tests for women at risk, follow-up and treatment of the affected women, early diagnosis of congenital Chagas disease in the newborn, and consequent early treatment.

### 2.2. Screening and Management for Pregnant Women

According to the Tuscany program, women “at risk” should be ideally identified by midwives during the first antenatal clinic consultation, when all pregnant women are given a small booklet that resumes the list of prescribed exams to be performed during pregnancy, according to National Italian law (or later on, during any moment of access to the healthcare system); of note, the pamphlet does not include the prescription for Chagas disease serology.

When taken in charge by the health care system at the beginning of a pregnancy, an information module (in Italian, Spanish and Portuguese versions) is given to the woman, containing information about the disease and its possible implications during pregnancy, how the serological screening can be performed, and how they will be looked after in case of positivity.

After counseling with the attending midwife, women at risk need to obtain a prescription for a Chagas disease serology test from their general practitioner, gynecologist, or another medical doctor in charge. With the prescription, women can perform the serology in any outpatient blood sampling structure in the Region along with the other blood tests recommended during pregnancy. Serum samples are sent to the Microbiology and Virology Unit, Careggi University Hospital, where serological tests are available.

Currently, a recombinant antigen-based serological test (Chemiluminescent Microparticle Immunoassay ARCHITECT Chagas^®^ (CMIA) from Abbott Laboratories, Wiesbaden, Germany) is used as a screening test. In the case of a positive serological test, a second test based on crude antigens (Ortho *T. cruzi* enzyme-linked immunosorbent assay (ELISA), from Ortho Clinical Diagnostics, Raritan, NJ, USA) is automatically performed to confirm the result. In case of discordant results, a third immunoblot test, which uses an exoantigen fraction from trypomastigote forms of *T. cruzi* (Chagas Western Blot IgG Assay^®^, LDBio Diagnostics, Lyon, France), is performed.

In the case of a confirmed diagnosis of Chagas disease, the woman will be actively re-called by the staff of the Regional Reference Center for Infectious Diseases in Pregnancy, and the Regional Reference Center for Tropical Diseases, both located in Careggi University Hospital. The women diagnosed with Chagas disease will undergo a clinical evaluation, including an assessment for organ damage, a 12-lead electrocardiography, an echocardiography, and an inquiry on possible gastrointestinal disease (presence of dysphagia or constipation). They will be counseled about the risk of congenital transmission and the diagnostic protocol that will follow after the neonate birth. The affected woman is also informed that vaginal delivery and breastfeeding are usually not contraindicated.

Parasitological treatment for Chagas disease will be offered after the end of breastfeeding; the same pathway will also be provided in the case of positive serology in women who do not carry the pregnancy to term.

The flow chart indicating the correct management of a pregnant woman at risk is illustrated in Figure 1. In case of Chagas disease diagnosis, treatment with benznidazole (or nifurtimox in case of intolerance) will be offered, according to WHO recommendations [3].

### 2.3. Newborns Screening, Management, and Follow-Up

Neonates born from mothers with Chagas disease will undergo clinical and laboratory evaluation right after delivery in order to exclude alarm signs for acute congenital Chagas disease, which include: overall impairment (Apgar score < 5 at 1′, <7 at 5′, weight < 2500 g, fever > 37.5 °C, hypothermia < 35 °C, lymphadenopathy, hepatosplenomegaly, jaundice, petechiae, edema or anasarca), signs of meningoencephalitis, myocarditis, respiratory distress, altered laboratory tests [24]. If alarm signs are detected, thick and thin blood smears and a third sample for real-time polymerase chain reaction (CHAG-UX, Progenie Molecular, Spain) for *T. cruzi* test are collected within the first 48 h after delivery. The mode of shipping and conservation of the specimens are specified in the protocol. If the newborn is asymptomatic, they will be re-evaluated at the Infectious _Disease Department in Meyer Pediatric Hospital, Florence, after 30–40 days of life. The complete flowchart established by the protocol is illustrated below (Figure 2).

For the newborns for whom congenital Chagas disease is confirmed and treatment is started according to WHO recommendations [3], the protocol provides precise guidance about routine follow-up and microbiological laboratory tests, and indications for instrumental examinations, timing, and interpretation of parasitological and serological tests.

If the mother had other children before the diagnosis, they would be followed up at Meyer Pediatric Hospital, following the same pathway that has been described for newborns.

### 2.4. Implementation Tools to Increase the Screening Coverage

In order to improve the implementation of the screening program, in November 2019, a “Chagas checklist” was introduced at Careggi University Hospital. It consists of a questionnaire that appears on the digital medical record and must be filled out when a pregnant woman is taken in charge by obstetricians and gynecologists upon admission for delivery. The checklist includes two questions:-“Was the pregnant woman born in one of the 21 endemic countries for Chagas disease?”

If the answer is “YES”, *T. cruzi* serology must be performed. If the answer is “NO”, the second question appears.

-“Was the mother of the pregnant woman born in one of the 21 endemic countries for Chagas disease?”

Again, if the answer is “YES” the serological test for *T. cruzi* must be performed.

Moreover, starting in 2019, several sections of the “Chagas disease in pregnancy” course were held, being addressed to all healthcare providers in the Tuscany region. The purpose of this course is to describe the main characteristics of Chagas disease and raise awareness about it, focusing on the disease during pregnancy and congenital transmission and illustrating the main points of the regional protocol for Chagas disease screening during pregnancy. The course is currently available online [25].

### 2.5. Implementation of Surveillance and Screening Outcomes

In order to analyze the screening coverage in Careggi University Hospital between 2012 and June 2022, information about the number of deliveries, country of origin of the parturient women and their mothers, completeness, and correctness of the “Chagas checklist” (when available, after the end of 2019) was obtained consulting the hospital’s archive and digital clinical charts, the latter available starting from 2017. In case of failure to test for Chagas disease in women at risk who gave birth after the introduction of the checklist in 2019, further consultation of the clinical chart was conducted in order to understand any reason for not performing the test.

Data about the number, type of test, and results of serological tests performed on at-risk women who gave birth in Careggi University Hospital were obtained by consulting the archive of the laboratory of Microbiology and Virology Unit. The annual screening coverage was calculated as the ratio between the number of tests performed on women at risk and the total number of deliveries at risk per year. Ethical approval was not required since the work is based on data collected for the monitoring of an approved Regional program.

## 3. Results

During the study period, 34,374 deliveries took place in Careggi University Hospital with an annual average of 3274. Among the women who gave birth between 2012 and June 2022, 1338 (3.9%) deliveries were from women who were born, or whose mother was born, in an endemic country for Chagas disease. The most represented country of origin was Peru, with 766 (57% of the total amount of women at risk) deliveries, followed by Brazil, with 239 (18%). Figure 3 shows the percentages of deliveries by women at risk according to country of origin.

Out of 1338 deliveries by women at risk, 626 (47%) *T. cruzi* serology tests were performed on women who gave birth in our center between 2012 and June 2022. Specifically, among tested women, four of them were born in Italy, but were considered at risk because they were born to a mother coming from a Latin American endemic country. Figure 4 compares the total number of deliveries by women at risk for Chagas disease with the number of deliveries by women at risk who were tested for Chagas disease.

As shown in Figure 5, on average, tests were performed in 50% of deliveries by women at risk each year; of note, the screening coverage progressively increased during the years, with an average screening coverage of 40.3% between 2012 and 2019, which arose to 75.7% between 2020 and June 2022. In particular, a significant increase in terms of the percentage of tested at-risk women was shown after the end of 2019, when the “Chagas checklist” was introduced in the digital charts system and the “Chagas disease in pregnancy” course was held for the first time.

Digital medical records of women who gave birth in Careggi University Hospital, compiled after the introduction of the “Chagas checklist” in November 2019, were checked in order to assess if and why any woman at risk was not tested. It emerged that 68 women who gave birth in our center were not tested, even though they were at risk, due to an error in filling the checklist.

There were 4 positive *T. cruzi* serology tests out of the 626 deliveries in which the test was performed (0.6%), corresponding to 2 affected women. It was not possible to calculate the seroprevalence among tested women at risk, because data about the number of deliveries by each woman, therefore the actual number of tested women, was available only after the introduction of the digital clinical chart in 2017. One woman who tested positive was born in the rural area of Chaco, Bolivia, and had three children in 2008, 2012, and 2013 respectively. All the children completed the follow-up period in accordance with the protocol and none of them were infected. The mother had been treated with benznidazole in 2008 for one month without completing the 2-month scheduled treatment because of several trips back to her country of origin. She never reported any symptom suggestive of cardiac or digestive clinical forms of Chagas disease. The second case was a woman from Chile who tested positive in 2015 and besides the newborn, she had another 5-year-old child at the time of diagnosis; both children were tested for congenital Chagas disease and the newborn completed the 9-month follow-up. Congenital infection was excluded for both of them. The mother was taken in charge at the infectious and tropical diseases unit of our center: electrocardiography, echocardiography, and gastrointestinal X-rays were normal, and she was asymptomatic. She was treated with benznidazole after the term of lactation.

## 4. Discussion

Chagas disease has been reported as the third most common imported Neglected Tropical Disease (NTD) in Italy, after strongyloidiasis and schistosomiasis [26]. Some steps towards the aim of improving the detection and treatment of most common NTDs, including Chagas disease, have been made in the past years in Italy, such as the diffusion of an expert consensus recommendation document for screening in immunocompromised migrants and the institution of national screening programs for blood and organ donors at risk [26,27,28,29]. However, only a few of these recommendations are followed in daily practice and national policies aimed at congenital Chagas disease prevention are lacking at the national level [20,26,27,28,29].

Even though other local initiatives exist in Italy [30,31], Tuscany is the first Region to introduce free serological screening for *T. cruzi* antibodies in pregnant women at risk and to offer a structured, detailed protocol to manage the affected mother and the newborn. However, the present study shows that screening coverage has only reached “sufficient” levels in recent years in Careggi University Hospital. Similar data come from the province of Alicante, where tests were performed in 39.8% of deliveries by women at risk between 2014 and 2018 despite the regional recommendation of universal screening for Chagas disease during pregnancy in women coming from endemic countries [32]. Of note, the main reasons for low screening coverage in this case were attributed to forgetfulness errors by midwives and obstetricians and lack of professional training on the screening program, together with a lack of knowledge of CD and of the possibility of its vertical transmission [32].

Even though data are not available, considering that Careggi University Hospital was the pioneer center promoting the screening protocol, we can presume that the screening coverage is even lower in other areas of Tuscany. It is worth mentioning that not only a large Latin American community is settled in Tuscany, of which 61.3% is composed of women, but also that available data from previous studies in our center indicate that 63.2% of people affected by Chagas disease are women, with a mean age of 39 [33]. This scenario evidences the need to reinforce the program at a regional level.

A reason for the low implementation level is that serology for *T. cruzi* can be performed upon an additional medical request as it is not included in the list of medical requests recommended by the Italian national health system. Moreover, the initiative is not supported by dedicated funds or personnel for promotional, educational, or training initiatives. The regional program, in fact, only covers the cost of the screening test and further investigations and the follow-up for positive women and their children. On the other hand, the scarce sensitivity towards the disease could be explained by a lack of knowledge and awareness of the disease among healthcare workers. This is, in fact, a common problem in nonendemic countries that is even worsened by the perception of a low disease rate in nonendemic areas [34,35]. The low frequency of Chagas disease diagnosis reflects, however, the need to implement screening programs, which would actually bring to light the real burden of the problem rather than a negligible healthcare issue [32]. Recent estimates pointed out that the prevalence of congenital Chagas disease at birth in children born from a Latin American mother in Italy is about 1:1670. This prevalence is higher than those of certain congenital diseases such as phenylketonuria (prevalence at birth 1:4500) and cystic fibrosis (prevalence at birth 1:5510), already included in the ”extended neonatal screening” program provided by Italian law since 1992 [36]. Moreover, based on Italian epidemiological statistics, seroprevalence in Latin American countries, and likely congenital transmission rate, it has been estimated that between 2014 and 2018, 463 (95% CI 267–792) women with Chagas disease gave birth and 16 (95% CI 12–21) congenital transmissions could have occurred [37]. However, only three cases of congenital transmission have been reported in the same period, two corresponding to the same family cluster and being diagnosed in the context of a screening program aimed at Latin Americans living in Milan [38,39,40]. Analogously, 2154 births are estimated to take place annually in the United States (US) from women affected by *T. cruzi* infection, with 22 to 108 consequently affected infants. However, besides two confirmed cases in infants born to immigrant mothers, there are no reports of congenital Chagas disease in the US [13,41,42]. Similar data come from Spain, where it was estimated that, between 2004 and 2018, 613 cases of Chagas infection occurred among children aged < 14 years old, 506 of them being born in Spain to Latin American mothers, with an estimated underdiagnosis of 60% [43].

These data suggest that worldwide implementation of health care policies in nonendemic countries is necessary to contain the perpetuation of Chagas disease transmission from one generation to the other. In the case of our study, the introduction of a mandatory checklist integrated within the digital medical record and the offering of online and onsite training courses has been a key driver for the success of the program. Indeed, the screening coverage significantly increased (from 40.3% between 2012 and 2019 to 75.7% from 2020 to June 2022) after the introduction of the two aforementioned implementation tools at the end of 2019. For these reasons, we think these strategies could represent a cheap and effective instrument that should be considered in any setting where this kind of screening can be implemented.

In this perspective, communication and cooperation between healthcare workers and policymakers are fundamental, and only with solid regional and national programs aimed at reducing congenital transmission, such as those implemented for screening in blood and organ donors, could the disease be properly controlled [8,29]. Thus far, only local and regional screening programs have been promoted; while Valencia, Galicia, and Catalonia, together with Tuscany, are the only realities where health policies have been adopted to control the congenital transmission of Chagas disease [23,44,45]. Other initiatives for congenital Chagas disease screening have been implemented in Bergamo (Italy) and Switzerland [30,46]. Screening programs for congenital Chagas disease prevention have also been promoted at the institutional level in Negrar (Verona), Rome, and Bologna; however, there are no published studies with data from these programs [31]. Besides these isolated initiatives, an example of a European or national program is still missing [8,9].

The relatively low number of cases of Chagas disease recorded in the study is partly explained by the fact that a large number of screened women were from Peru, where the disease prevalence ranges around 0.7% [4]. These results, together with other epidemiological results on Chagas disease in Italy, enforce the need to adopt a homogeneous approach to this healthcare issue in order to get reliable epidemiological data about the seroprevalence of Chagas disease in Tuscany and the Italian country and to adopt adequate measures for prevention of transmission and easier access to diagnosis and treatment of the disease [17,26,47].

Moreover, it has been demonstrated that not only screening programs for congenital transmission and early treatment of the newborn are strongly cost-effective, but also screening programs for Chagas disease in the general adult population are cost-effective if compared with the costs of diagnosis, management, support, and treatment of the chronic complications of the disease [48,49]. Analogously, widespread treatment of infected women of childbearing age has been demonstrated to be strongly effective in controlling and stopping mother-to-child transmission [50]. As a matter of fact, since 2018, the WHO strategy has shifted from control to prevention and elimination of congenital transmission of Chagas diseases, following growing evidence demonstrating that treating infected women of childbearing age before pregnancy can effectively prevent congenital transmission [51,52]. Ideally, programs for congenital Chagas disease should include not only testing of pregnant women at risk and follow-up of neonates born from infected mothers but also pre-conceptional testing of women of childbearing age at risk age and treatment of infected mothers before pregnancy [53].

From this perspective, it is worth noting that, in June 2022, Unitaid and the Pan American Health Organization launched a five-year partnership that aims to eliminate mother-to-child transmission of Chagas disease through regional and national efforts. Among the main objectives of the project, they included the determination of the efficacy of new shorter regimens for Chagas disease [6]. In fact, with the phase 2 trial by Torrico et al., the efficacy of shorter benznidazole treatment regimens has been demonstrated for the first time and deserves further studies since the implementation of shorter treatments could substantially improve the tolerability and accessibility of the drug [54].

Moreover, it should be taken into account that treatment for Chagas disease usually has 100% efficacy in the newborn; this value tends to decrease progressively as the chronic disease is prolonged [55]. Therefore, early detection of congenital Chagas disease would not only guarantee the possibility of a simple therapeutic intervention on the newborn, which would be well-tolerated and effective but would also make it possible to cut the subsequent costs related to the care of the complications themselves [56].

One limitation of this work is that collected data cover only information about Careggi University Hospital and do not include all other hospitals in the Region, due to the fact that different electronic systems are used in different hospitals and electronic systems cannot communicate with each other. A homogenized surveillance system is needed in order to get reliable epidemiological data at a regional level. We also point out that PCR for *T. cruzi* was provided by the protocol, together with an hemoscopic exam for parasitological direct diagnosis; however, molecular methods are not standardized for the diagnosis of Chagas disease and only a few reports evaluated the performance of commercial methods [57,58]. In order to promote the use of these high-sensitivity tests, they should be standardized and there should be a consensus by the scientific community on their use [59,60]. Finally, even though the Chagas checklist and dedicated training courses on the topic were demonstrated to be good strategies for the implementation of the regional program, results can still be improved in terms of screening coverage. Some strategies could rely on awareness raising through dedicated public events and professional training courses could be strengthened and repeated over time. Moreover, if Chagas disease serology could be included in the pregnancy pamphlet among all the other exams prescribed during physiological pregnancy, it would be easier for midwives and obstetricians to remember to perform it when appropriate.

## 5. Conclusions

The screening program for congenital Chagas disease in Tuscany needs to be strengthened to better reach all the areas of the Region. There is a big need to raise awareness of this neglected disease among all healthcare providers and not only among infectious diseases specialists, because only through communication and collaboration between different actors of the healthcare system can the real burden of the disease be measured and controlled with adequate health policies. This work wants to encourage the implementation of screening and surveillance programs for congenital Chagas disease at a national and European level. In this perspective, the introduction of cheap but strongly effective implementation strategies, such as integrated checklists and training courses, could contribute to the aim of reaching better awareness of the epidemiological features of the disease in Europe and of controlling the spread of Chagas disease in the world.

## Figures and Tables

**Figure 1 microorganisms-11-00154-f001:**
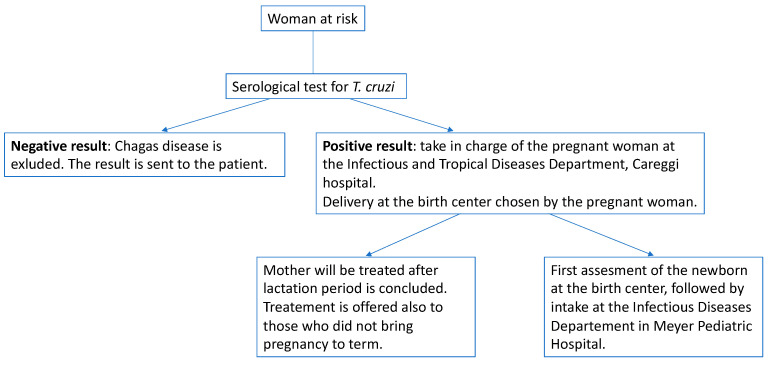
Steps provided by the protocol when a pregnant woman at risk is taken in charge.

**Figure 2 microorganisms-11-00154-f002:**
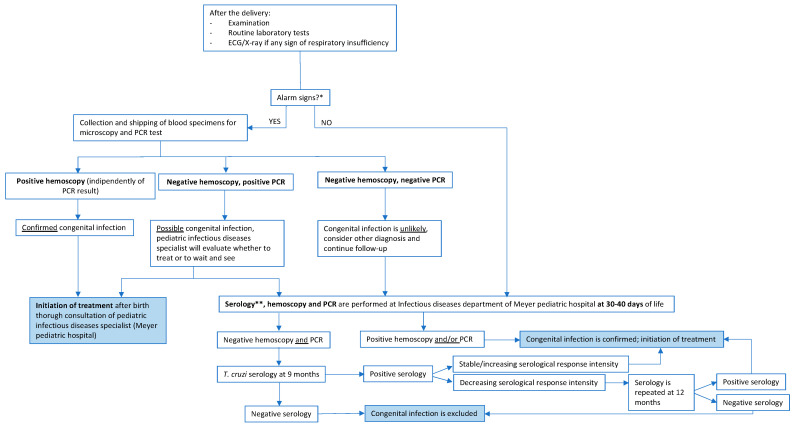
Scenarios and management of the infant born to a mother with Chagas disease. * Alarm signs include: overall impairment (Apgar score < 5 at 1′, <7 at 5′, weight < 2500 g, fever > 37.5 °C, hypothermia < 35 °C, lymphadenopathy, hepatosplenomegaly, jaundice, petechiae, edema or anasarca), signs of meningoencephalitis, myocarditis, respiratory distress, altered laboratory tests ** Serology performed at one month will be positive even in the absence of congenital infection, by passive transfer of maternal IgG antibodies. However, it is useful to make a comparison with the serology that will be performed at 9 months for a better interpretation of the result.

**Figure 3 microorganisms-11-00154-f003:**
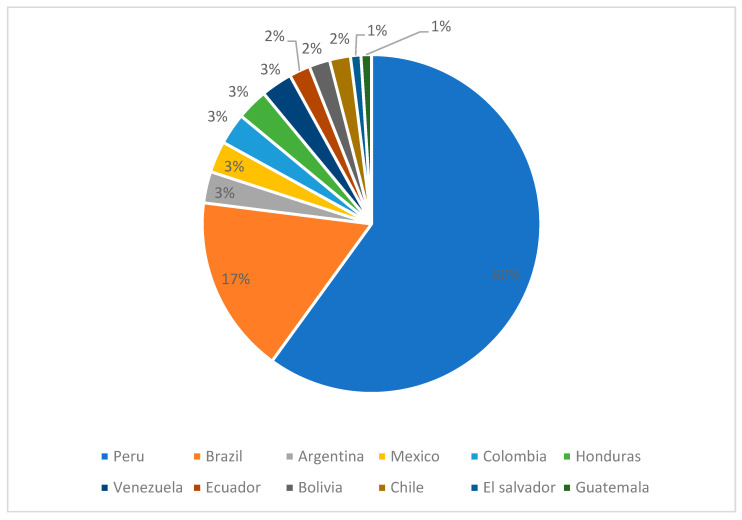
Percentages of deliveries (total N = 1338) by women at risk who gave birth in Careggi University Hospital, according to country of origin.

**Figure 4 microorganisms-11-00154-f004:**
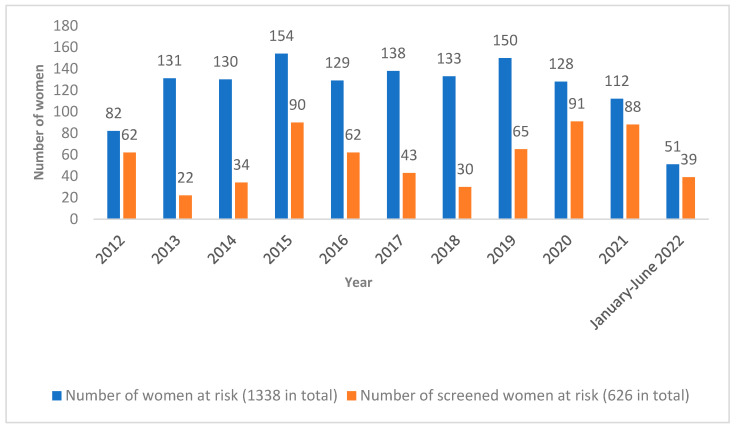
Representation of the total number of deliveries by women at risk per year (blue), and total number of women at risk tested for Chagas disease who gave birth in Careggi University Hospital each year (orange).

**Figure 5 microorganisms-11-00154-f005:**
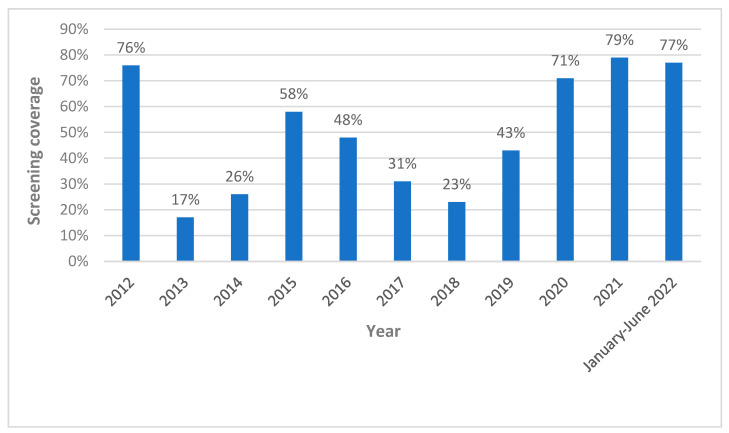
Percentage of Chagas disease screening coverage per year in women at risk who gave birth in Careggi University Hospital.

## Data Availability

Not applicable.

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
