# Peer review of "A Questionnaire Integrated with the Digital Medical Record Improved the Coverage of a Control Program for Congenital Chagas Disease in Tuscany, Italy"

_microorganisms, 2023, doi:10.3390/microorganisms11010154_

Round 1
Reviewer 1 Report
The aim of this work is important given the lack of accuracy in the diagnosis of Chagas disease (In LATAM and Europe) and, above all, the morbidity rates that occur in patients. The relevance of screening in European countries in pregnant women highlights the importance of this disease, in addition to avoiding fatal outcomes in children who have been infected from the mother.
In line 59, change "continental Latin America" ​​to Latin American countries.
Question:
1) Why didn't you consider the use of q-PCR in peripheral blood of pregnant women?
2) Thinkin about the biology of T. cruzi, it is known that depending on the region there is a different distribution of DTUs, there are reports that indicate that a serological test can work in a country according to the T. cruzi DTU´s used for the serological test but in other countries it does not have the same effectiveness.
What was your criteria for the selection of the serological tests used for tha diagnosis?
Do you think there were false positive results?
Reviewer 2 Report
Comments and suggestions for the authors are included in the attached word file.
